# Record of the Emerald Ash Borer (*Agrilus planipennis*) in Ukraine is Confirmed

**DOI:** 10.3390/insects10100338

**Published:** 2019-10-11

**Authors:** Alexander N. Drogvalenko, Marina J. Orlova-Bienkowskaja, Andrzej O. Bieńkowski

**Affiliations:** 1V.N. Karazin Kharkiv National University, 4 Svobody sq., 61022 Kharkiv, Ukraine; triplaxxx@ukr.net; 2A.N. Severtsov Institute of Ecology and Evolution of Russian Academy of Sciences, 119071 Moscow, Russia; bienkowski@yandex.ru

**Keywords:** emerald ash borer, EAB, Ukraine, Europe, *Fraxinus pennsylvanica*, ash trees, invasive pest, plant quarantine

## Abstract

*Agrilus planipennis* (Coleoptera: Buprestidae) is a devastating invasive pest of ash trees. This wood-boring insect is native to Asia and established in European Russia about 20 years ago. It severely damages *Fraxinus pennsylvanica* plantations and quickly spreads. In 2019 we first detected *A. planipennis* in Ukraine. More than 20 larvae were collected from under the bark of *F. pennsylvanica* trees on 5 September 2019 in the Markivka District of the Luhansk Region. The coordinates of the localities of collection were 49.614991 N, 39.559743 E; 49.614160 N, 39.572402 E; and 49.597043 N, 39.561811 E. The photos of the damaged trees with larval galleries, exit holes and larvae are presented. It indicates that *A. planipennis* is established in the east of Ukraine. This fact is important for development of quarantine protocols to prevent or at least slow the further spread of this invasive pest in Europe.

## 1. Introduction

The emerald ash borer (EAB), *Agrilus planipennis* (Coleoptera: Buprestidae), a pest of ash trees (*Fraxinus* spp.), is native to China, Russian Far East, Japan and Korea [1]. Since its accidental introduction to North America in the 1990s, this devastating pest has widely spread in Canada and the United States and killed hundreds of millions of ash trees [2]. Similarly, the introduction of EAB into Moscow, again back in the 1990s, has likewise led to widespread mortality of *F. pennsylvanica* in an ever-expanding outward range [3,4]. The spread of *A. planipennis* should be carefully monitored, because it poses a serious threat to *F. pennsylvanica* plantations in Europe [5]. Before 2019 *A. planipennis* was recorded only in European Russia. In 2018, it was recorded in the very south of the Voronezh Region of Russia, i.e. near the border of the Luhansk Region of Ukraine [6]. Information about the spread *A. planipennis* into another country is crucial for plant quarantine protocols. Thus, we decided to look for the pest in the Luhansk Region of Ukraine.

## 2. Materials and Methods 

Between 20 and 22 June 2019 ash trees in the Starokozhiv Forest and the field shelter belt in its vicinity (the Markivka District of the Luhansk Region of Ukraine) were examined by A.N. Drogvalenko. This locality was chosen for the survey because it is just about 25 km from the nearest known *A. planipennis* locality in Russia [6]. The stems of about 250 ash trees (*F. excelsior* and *F. pennsylvanica*) were examined for the characteristic D-shaped exit holes. Three trees of *F. pennsylvanica* damaged by *A. planipennis* were detected. These trees were situated at the edge of the forest belts and had a diameter of 7–10 cm. Characteristic D-shaped exit holes were situated at a height of 50–200 cm. The infested trees had dying of upper branches, reduced foliage density (small leaves) and fewer seeds. This information was included to the paper posted as a preprint to bioRxiv on 2 July 2019 [7].

Immediately following the appearance of this preprint on the Internet, the National Plant Protection Organization of Ukraine conducted an official survey in the same area and did not detect *A. planipennis*. And since we had no specimens or photos for confirmation, our record of *A. planipennis* in Ukraine was considered unreliable [8]. A description of the forest provided in the report of the Ukrainian Plant Protection Organization indicates that the employees of this organization did not find the infested forest belt. Unfortunately, they did not ask A.N. Drogvalenko to show them these trees or indicate the exact coordinates of his finding.

Between 4 and 6 September 2019 A.N. Drogvalenko visited the Markivka District of the Luhansk Region of Ukraine again and repeated the survey of the ash trees. His aim was to take photos of the exit holes, larval galleries and larvae of *A. planipennis*, and to collect larvae from under the bark. Larvae were identified by an illustrated guide to distinguish emerald ash borer (*A. planipennis*) from its congeners in Europe [9].

## 3. Results

The same three infested trees and more than 40 other *F. pennsylvanica* trees, heavily infested with *A. planipennis*, were found in this region (Figure 1). More than 20 larvae were collected and preserved in alcohol. The larvae under the bark were found to be of different instars including the last instar. The coordinates of the trees where the larvae were collected are 49.614991 N, 39.559743 E; 49.614160 N, 39.572402 E; and 49.597043 N, 39.561811 E (roadside plantation). The larvae are in the collection by A.N. Drogvalenko in Kharkiv.

No infestations of *F. excelsior* were detected in spite of this ash species being common in this forest.

## 4. Discussion

Since the examined trees are heavily infested and some larvae are of the last instar, it is obvious that the infestation is at least two years old. The finding of EAB in Ukraine is not surprising. By 2019 EAB has spread to 14 regions of European Russia: Bryansk, Kaluga, Lipetsk, Moscow, Orel, Ryazan, Smolensk, Tambov, Tula, Tver, Vladimir, Volgograd, Voronezh and Yaroslavl [7]. The distance between the entry point of invasion (Moscow) and the most remote known EAB locality (the officially declared plant quarantine zone in Volgograd [10]) is about 900 km. The distance from Moscow to the locality of the EAB detection in Ukraine is about 700 km. Therefore, it is not excluded that EAB is already widespread in Ukraine.

## 5. Conclusions

There is no doubt that *A. planipennis* is established in Ukraine. It should be taken into account in plant quarantine protocols of European countries.

## Figures and Tables

**Figure 1 insects-10-00338-f001:**
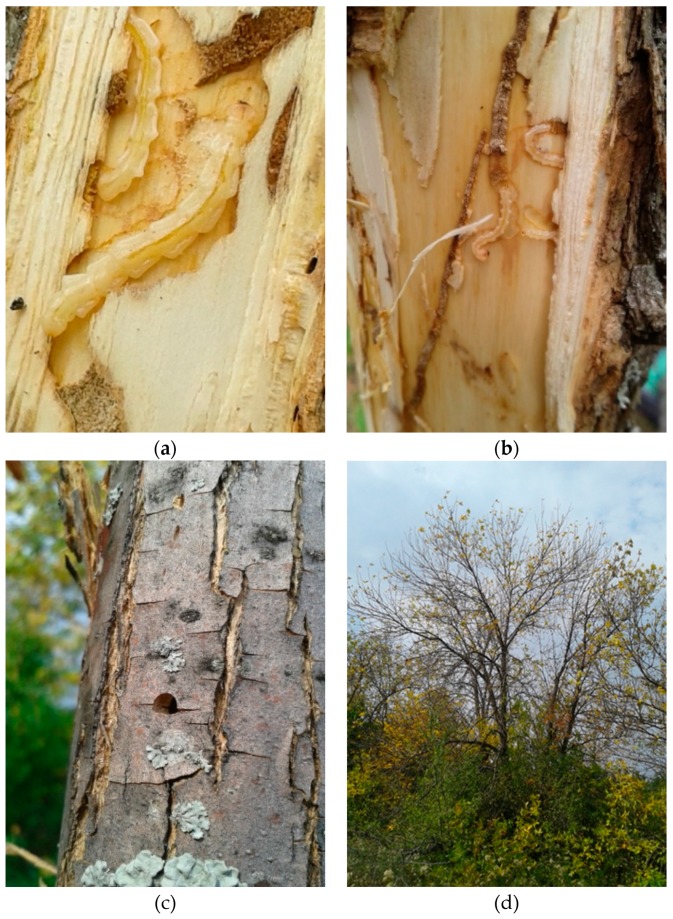
Symptoms of emerald ash borer (EAB) infestation on ash trees (*Fraxinus pennsylvanica*) in the Markivka Disrtict of Ukraine. (**a**,**b**) Larvae of *A. planipennis*; (**c**) exit hole; and (**d**) a damaged tree.

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
