# Peer review of "Record of the Emerald Ash Borer (Agrilus planipennis) in Ukraine is Confirmed"

_insects, 2019, doi:10.3390/insects10100338_

Round 1
Reviewer 1 Report
This is a very brief, succinct and well written account of a new country record for an important invasive insect.
It is worthy of publishing just to establish this in the scientific literature.
I found no problems with the manuscript.
Author Response
Thank you for this review.
Reviewer 2 Report
This short communication depicts the first record of Agrilus planipennis in the East Ukraine. This beetle species is invasive to parts of North America and causes significant ash diebacks. The record is interesting to a readership of forest entomologists. However, I have some comments, that might help to improve the manuscript.
First, the text is a bit sensationally. I suggest rephrasing it with some caution. E.g.:
12- the authors do not have data from all over Europe. So please rephrase to “possible threat to European ash trees” – especially since the records where made in F. pennsylvanica not F. excelsior.
16 – it’s a hint that there is reproduction of A. planipennis. So may rephrase to “…indicate that A. planipennis is established in the Ukraine.”
Do not refer to “the pest” in the entire text, but use the correct scientific species name.
Second, there seem to be several minor flaws in the language, especially with using articles etc. I suggest the authors to make use of some professional language editing service or the help of an English native speaking peer.
Finally, the authors might be interested in adding a small discussion of if Fraxinus excelsior occur in the same area and if there is also infestation.
Author Response
Thank you very much for the quick review and valuable comments.
12- the authors do not have data from all over Europe. So please rephrase to “possible threat to European ash trees” – especially since the records where made in F. pennsylvanica not F. excelsior.
- Yes. We rephrased it in the Abstract and in the Introduction.: "It severely damages Fraxinus pennsylvanica plantations."
16 – it’s a hint that there is reproduction of A. planipennis. So may rephrase to “…indicate that A. planipennis is established in the Ukraine.”
Yes. It is rephrased.
Do not refer to “the pest” in the entire text, but use the correct scientific species name.
- Yes. “The pest” is replaced with "A. planipennis".
Second, there seem to be several minor flaws in the language, especially with using articles etc. I suggest the authors to make use of some professional language editing service or the help of an English native speaking peer.
- The help of the language editing service would be very useful. We don’t have enough time for this. The deadline for the upload of the revised manuscript is October 8.
Finally, the authors might be interested in adding a small discussion of if Fraxinus excelsior occur in the same area and if there is also infestation.
- Yes. It is added: "No infestations of Fraxinus excelsior are detected in spite of this ash species is usual in this forest."
Reviewer 3 Report
This short paper is meant to re-affirm the presence of the EAB in the Ukraine.
How do the authors know that the larvae belong to the EAB and not to another buprestid species? In the Introduction should be mentioned if other buprestids are known to attack ash trees in the Ukraine and in the Results to provide some features of the larvae that establish that they are EAB larvae.
While many entomologists would know that Agrilus is a buprestid, others wouldn’t so I suggest placing “(Coleoptera: Buprestidae)” in the title and certainly in the abstract.
Author Response
Thank you for the quick review and valuable comments.
How do the authors know that the larvae belong to the EAB and not to another buprestid species? In the Introduction should be mentioned if other buprestids are known to attack ash trees in the Ukraine and in the Results to provide some features of the larvae that establish that they are EAB larvae.
- Yes. Thank you. To clarify this we have added the sentence: "Larvae of Agrilus planipennis were identified with the help of an Illustrated guide to distinguish emerald ash borer (Agrilus planipennis) from its congeners in Europe." We added the reference: Volkovitsh, M.G.; Orlova-Bienkowskaja, M.J.; Kovalev, A.V.; Bieńkowski, A.O. An illustrated guide to distinguish emerald ash borer (Agrilus planipennis) from its congeners in Europe. Forestry 2019. DOI:10.1093/forestry/cpz024
While many entomologists would know that Agrilus is a buprestid, others wouldn’t so I suggest placing “(Coleoptera: Buprestidae)” in the title and certainly in the abstract.
- Yes. It is added.